# Multidrug-Resistant Gram-Negative Bacteria Decolonization in Immunocompromised Patients: A Focus on Fecal Microbiota Transplantation

**DOI:** 10.3390/ijms21165619

**Published:** 2020-08-05

**Authors:** Laura Alagna, Emanuele Palomba, Davide Mangioni, Giorgio Bozzi, Andrea Lombardi, Riccardo Ungaro, Valeria Castelli, Daniele Prati, Maurizio Vecchi, Antonio Muscatello, Alessandra Bandera, Andrea Gori

**Affiliations:** 1Infectious Disease Unit, Fondazione IRCCS Ca’ Granda Ospedale Maggiore Policlinico, 20122 Milan, Italy; laura.alagna@policlinico.mi.it (L.A.); davide.mangioni@unimi.it (D.M.); giorgio.bozzi@policlinico.mi.it (G.B.); andrea.lombardi@policlinico.mi.it (A.L.); riccardo.ungaro@unimi.it (R.U.); valeria.castelli@unimi.it (V.C.); antonio.muscatello@policlinico.mi.it (A.M.); alessandra.bandera@unimi.it (A.B.); andrea.gori@unimi.it (A.G.); 2Department of Pathophysiology and Transplantation, University of Milan, 20122 Milan, Italy; maurizio.vecchi@unimi.it; 3Centre for Multidisciplinary Research in Health Science, University of Milan, 20122 Milan, Italy; 4Department of Medical Biotechnology and Translational Medicine, University of Milan, 20122 Milan, Italy; 5Department of Transfusion Medicine and Hematology, Fondazione IRCCS Ca’ Granda Ospedale Maggiore Policlinico, 20122 Milan, Italy; daniele.prati@policlinico.mi.it; 6Gastroenterology and Endoscopy Unit, Fondazione IRCCS Ca’ Granda Ospedale Maggiore Policlinico, 20122 Milan, Italy

**Keywords:** gut colonization, decolonization, fecal microbiota transplantation, antimicrobial resistance, multidrug-resistant, immunocompromised, gut microbiota, dysbiosis

## Abstract

Antimicrobial resistance is an important issue for global health; in immunocompromised patients, such as solid organ and hematological transplant recipients, it poses an even bigger threat. Colonization by multidrug-resistant (MDR) bacteria was acknowledged as a strong risk factor to subsequent infections, especially in individuals with a compromised immune system. A growing pile of studies has linked the imbalance caused by the dominance of certain taxa populating the gut, also known as intestinal microbiota dysbiosis, to an increased risk of MDR bacteria colonization. Several attempts were proposed to modulate the gut microbiota. Particularly, fecal microbiota transplantation (FMT) was successfully applied to treat conditions like *Clostridioides difficile* infection and other diseases linked to gut microbiota dysbiosis. In this review we aimed to provide a look at the data gathered so far on FMT, focusing on its possible role in treating MDR colonization in the setting of immunocompromised patients and analyzing its efficacy and safety.

## 1. Burden of Multi-Drug Resistant (MDR) Bacteria

Antimicrobial resistance (AMR) is today one of the biggest threats to global health, requiring urgent containment measures. A significant increase in the number of infections and the deaths attributable to AMR was registered between 2007 and 2015 [1], with an increase of AMR prevalence in Italy from 1.3% in 2009 to 26.8% in 2018 (data available at https://atlas.ecdc.europa.eu/).

Currently, the most widely accepted definition of multidrug-resistant (MDR) bacteria include a lack of susceptibility in three or more antimicrobial categories active against the isolated microorganism [2]. In this review, we focused on MDR gram-negative bacteria (MDR-GNB) because of their significance in immunocompromised patients, and explored the strategies and management methods used to address the issue of colonization and subsequent infection.

Enterobacterales is a large family of Gram Negative (GN) bacteria that populate the intestinal microbiota of healthy hosts. Bacteria from this family are a common cause of severe infections, representing the most frequent cause of bloodstream infections (BSIs), urinary tract infections (UTIs), and one of the important causes of pneumonia both in community and healthcare settings. Among these bacteria, a major mechanism of AMR is the production of extended spectrum β-lactamases (ESBLs) and carbapenemases.

Data on AMR in Europe are reported by the European Antimicrobial Resistance Surveillance Network (EARS-Net, www.ecdc.europa.eu/en/about-us/partnerships-and-networks/disease-and-laboratory-networks/ears-net).

EARS-Net collects results from routine antimicrobial susceptibility testing of invasive isolates (blood and cerebrospinal fluid cultures) from clinical laboratories, in each of the 30 European countries involved. The use of EUCAST clinical breakpoints is encouraged.

Data from European Antimicrobial Resistance Surveillance Network (EARS-Net) 2018 report, stated that more than half (58.3%) of the *Escherichia coli* isolates responsible for invasive diseases were resistant to at least one of the antimicrobial groups under regular surveillance (i.e., aminopenicillins, fluoroquinolones, third-generation cephalosporins, aminoglycosides, and carbapenems). The highest population-weighted mean resistance percentage was reported for aminopenicillins (57.4%), followed by fluoroquinolones (25.3%), third-generation cephalosporins (15.1%), and aminoglycosides (11.1%), with a significant increasing trend. On the contrary, *E. coli* resistance to carbapenems remains rare in Europe, ranging from 0 to 2%. Italy is one of the European countries with the highest percentage of AMR: In 2017, 41.7% of *E. coli* isolated from blood and cerebrospinal fluid were resistant to fluoroquinolones, and 28.7% were resistant to third-generation cephalosporins [3].

Like *E. coli*, *Klebsiella pneumoniae* can also be resistant to multiple antimicrobial agents. According to the EARS-Net 2018 report, more than a third (37.2%) of *K. pneumoniae* isolates were resistant to at least one of the aforementioned antimicrobial groups.

The highest population-weighted mean resistance percentage was reported for third-generation cephalosporins (31.7%), followed by fluoroquinolones (31.6%), aminoglycosides (22.7%), and carbapenems (7.5%).

Percentages of invasive isolates of carbapenem-resistant *K. pneumoniae* (CR-Kp) in Europe show a large variability, ranging from 0 to 64.7% [4]. The population-weighted mean percentage varied between 2014 and 2017, and was 7.3% in 2014 and 7.2% in 2017.

In Italy, 53.6% of *K. pneumoniae* invasive isolates were resistant to third-generation cephalosporins and 26.8% to carbapenems [3].

## 2. Gut Microbiota Dysbiosis: A Favorable Environment for MDR Bacteria

The small and large bowel house one of the widest microbial communities in the human body, both in terms of microbial density and diversity—the gut microbiota [5]. Most of these species belong to the Bacteroidetes, Firmicutes, Actinobacteria, and to a lesser extent to the Proteobacteria phyla. However, the relative proportions of each taxon vary considerably between individuals and even within the same individual during their lifetime. The composition of an individual’s microbiota is influenced by many factors, ranging from age and geographical location to eating habits, comorbidities, and the use of probiotics, prebiotics, and antibiotics [6]. 

Measures borrowed from ecology are used to describe microbiome diversity—α-diversity is the variance within a particular sample (e.g., the mean diversity in one individual gut microbioma), whereas β-diversity represents how samples vary against each other (e.g., different ecosystems sampled within the same host or among distinct individuals). The balance within microbial communities has a crucial role in maintaining a healthy state. The disruption of the normal intestinal microbiota, known as dysbiosis, can adversely affect the host’s health, and critically, cause loss of protection against colonization.

In its eubiotic condition, the gut microbiota plays an important role endowing the host with resistance against a wide range of pathogens, a mechanism known as “colonization resistance” [7].

First, it enhances the development of the immune system and the maintenance of the epithelial barrier by involving molecules and signaling pathways, such as short-chain fatty acids (SCFA) and bile acids [8]. SCFAs, produced by bacteria through fermentation of non-digestible carbohydrates, can interfere with bacterial growth by lowering their intracellular pH and by affecting their metabolic functioning (e.g., prevention of synthesis of methionine and accumulation of toxic homocysteine in *E. coli* strands [9]). Bile acids, such as deoxycholic acid, can have a bactericidal function by provoking membrane disruption and subsequent leakage of cellular content; this secondary acid, produced mostly by the *Clostridium* species (namely *C. scindens*), has a known toxic effect on many microorganisms, including *Staphylococcus aureus* and *Clostridioides difficile* [10].

Secondly, the intestinal commensal bacteria compete directly with pathogen species by means of depletion of nutrients and secretion of bacteriocins. Bacteria, especially those belonging to the same species, have to compete for nutrients, and the presence of commensal strains can inhibit the colonization by pathogen strains [11]. This paradigm can be extended from macronutrients (such as a specific carbohydrate) to micronutrients (such as iron, an important asset for bacterial growth). Probiotic strains that are able to effectively scavenge iron indirectly inhibit their competitors’ capability of surviving [12]. Furthermore, bacteria can produce bacteriocins, toxic peptides that hamper the growth of other species, and their capability to colonize the intestine [13]. These molecules (such as nisin, thuricin, and colicin) have different mechanisms of action to affect bacterial growth and survival, ranging from disruption of nucleic acid metabolism to pore formation in the bacterial membrane. Bacteriocins often mimic the effect of antibiotics. Thuricin, produced by *Bacillus thuringiensis*, has proved to be as effective as metronidazole and vancomycin in targeting *C. difficile* pathogenic strains [14].

Lastly, the composition of intestinal microbiota is a contributing factor to the integrity of the mucus layer of the gut barrier, which represents the first physical defense against colonization by exogenous bacteria. A dietary fiber-deprived gut microbiota was linked to the thinning of such mucus layer and to an enhanced susceptibility to pathogenic colonization [15].

Therefore, intestinal microbiota dysbiosis represents an ecosystem that is a favorable reservoir of MDR bacteria, not only by means of facilitating colonization from outer microorganisms but also by enriching the collection of AMR genes that can be found within the bacteria colonizing the gastrointestinal tract, also known as the intestinal resistome [16,17]. Indeed, gut microbiota hosts thousands of bacterial species, including well-known pathogens (e.g., *Enterococcus* spp, Enterobacterales) that are capable of sharing AMR genes among themselves. Additionally, it was theorized that resistance genes can be acquired or donated even from/to microorganisms that are just passing through the intestine [18]. In their study, Salyers et al. proved that transfer of AMR genes occurs between Gram-positive and Gram-negative bacteria in the mammalian colon, finding virtually identical resistance genes in natural isolates representing different bacterial species. Ruppé et al. [19], using metagenomic sequencing, proposed that the majority of AMR determinants are harbored by commensal bacteria of the intestinal microbiota as an innate asset to preserve the microbiota itself from outer threats like antibiotic exposure.

Immunocompromised patients, such as solid organ and hematopoietic stem cell transplant recipients and individuals with blood disorders, show the poorest intestinal microbioma diversity [20]. In these patients, the use of chemotherapeutic agents, combined with frequent antibiotic use like prophylaxis or treatment, impairs the host immune system and gut microbiota eubiosis. Intestinal domination by Gram-negative bacteria, defined as occupation of at least 30% of the microbiota by a single predominating bacterial taxon, is relatively common. Allowing the translocation of pathogens into the bloodstream is a risk factor for life-threatening systemic infection, especially during episodes of neutropenia and damage to the gastrointestinal system [21]. More than 20 years ago, MacFie et al. [22] determined how gut microbiome colonization and translocation had a pathogenetic role in post-operative sepsis, comparing cultures of nasogastric aspirates and mesenteric lymph nodes in surgical patients, and noting how positive cultures for multiple organisms were linked to increased bacterial translocation and septic morbidity.

It was also postulated that significant changes in the intestinal microbiota composition happen during the course of intensive care unit (ICU) stay, causing a profound state of immunosuppression and increasing the risk of organ failure and mortality [23,24,25]. Shimizu et al. [26] investigated changes of the gut ecosystem in vivo within patients with systemic inflammatory response syndrome, detecting a severe imbalance in intestinal flora. The authors found reduction of probiotic anaerobic bacteria such as *Bifidobacterium* and *Lactobacillus*, which have a protective role by increasing SCFA concentrations, enhancing humoral immune response, and stimulating cellular immunity [27]. On the contrary, there was an overgrowth of potentially pathogenic *Staphylococcus* and *Pseudomonas* species.

The role of the intestinal microbiota in the host capability to face a pathogenic noxa was studied in animal models. A greater mean species diversity (α-diversity) among the gut microbial communities in mice was linked to a higher chance of survival in the experimental models of abdominal sepsis. Such increased survival was linked to an increased CD4+ T cell response [28]. In addition, the authors co-housed mice with different baseline α-diversity and showed increased sepsis survival and improvement in T cell response and IgA production in mice that were more likely to die prior to cohousing.

There is still little data on the impact of the intestinal microbiome on immune modulation in humans. Two large epidemiological studies addressed the correlation between the risk of sepsis and recurrence of *C. difficile* infection (CDI) and broad-spectrum antibiotics usage, considering the latter to be an epiphenomenon of gut microbiota dysbiosis [29,30]. Both studies concluded that there was a higher risk of developing sepsis in patients with altered intestinal diversity, and that antibiotics often associated with CDI (such as third- and fourth-generation cephalosporins, fluoroquinolones, and carbapenems) had a stronger association to sepsis and septic shock.

Administration of antibiotics can indeed impair colonization resistance, not only by altering the composition of taxa, but also by affecting gene expression, protein activity, and overall metabolism of the bacteria populating the gastro-intestinal tract [31]. By documenting and tracing the transmission of CR-Kp during an hospital outbreak, Snitkin et al. [32] proved that antibiotic treatment causes a marked expansion of Enterobacterales, which in turn contributes to the transmission of AMR strains from patient-to-patient within hospitals.

The most studied clinical example of the impact of antibiotics on gut microbiome is the CDI. This microorganism mainly causes infections in hospital patients and residents of long-term care facilities, mostly due to the use of broad-spectrum antibiotics. Although almost any antibiotic might increase the risk of developing CDI, this condition is usually associated with fluoroquinolones, cephalosporins, and mainly clindamycin, which is excreted in the bile and, therefore, reaches high concentrations in stools. In experimental models, a single dose of clindamycin significantly reduces the diversity of the gut microbiota [33].

## 3. Risk Factors for MDR Infections and Colonization—Clinical Point of View in Immunocompromised Patients

### 3.1. Solid Organ Transplant Recipients

In the last decades, the introduction of effective immunosuppressive agents has reduced the risk of rejection of transplanted organs; nonetheless, the long-term immunosuppressive state of solid-organ transplant (SOT) recipients has increased their susceptibility to infections. Bacterial infections currently represent a leading cause of morbidity and mortality among SOT recipients and threaten the chance of graft survival [34]. Moreover, chronic liver diseases leading to cirrhosis were linked to alteration in gut microbiome [35]. The severity of bacterial infections in SOT recipients is increased by the emergence of multidrug resistant strains and approximately 20% of SOT recipients have an infection by MDR bacteria [36]. Rate of morbidity and mortality associated with MDR-GNB, in particular carbapenem-resistant Enterobacterales (CRE), are significantly higher compared with infection sustained by carbapenem susceptible strains [37].

Different studies aimed to analyze the risk factors for MDR colonization or infections, and appropriate management of colonized patients among candidates of SOTs is still a debated topic. A more accurate definition of risk factors for colonization and infection is necessary to improve prevention and management strategies that are able to reduce the rate of infections and associated mortality.

According to the recent Spanish Transplantation Infection Study Group (GESITRA) recommendations, MDR colonization does not constitute a contraindication for SOT [38], nevertheless the safest approach to colonization management is yet to be defined.

To the best of our knowledge, different surgical prophylaxis regimens are not recommended for patients colonized with MDR-GNB, but empirical treatment that includes active antibiotics should be administered in case of infective episodes in SOT recipients that are known to be colonized [38].

Below, we summarized key data on risk factors associated with ESBL-producing Enterobacterales and CRE colonization/infection in SOT.

#### 3.1.1. Colonization/Infection by ESBL-Producing Enterobacterales

Infections by ESBL-producing Enterobacterales are more frequent among kidney transplant (KT) patients because the urinary tract is the source for most of the post-transplant infections, with infection rates ranging from 3 to 11% [39,40]. The associated mortality is 15–26% at 30 days after transplant [34,38].

Among KT patients, specific risk factors for infections by ESBL-producing bacteria were identified as—pancreas transplantation combined with KT, prior use of antibiotics, renal replacement therapy, and post-transplant urinary obstruction [39,40]. Of note, 55% of patients experiencing infections by ESBL-producing bacteria have a previous rectal colonization [41].

Similarly, infections by ESBL-producing bacteria are a concern in liver transplant (LT) patients. In this setting, the prevalence of infection was estimated to be 7% with a 15–26% 30-day mortality rate. In LT recipients, liver failure was identified as an independent risk factor for ESBL colonization, and additional risk factors for ESBL acquisition and colonization were multiple surgeries, previous exposure to antibiotics and intubation longer than 72 h [42].

At present, the impact of previous ESBL colonization on the incidence of infection needs to be assessed. Bert et al. [43] analyzed 710 LT patients, reporting that 44.8% of colonized recipients developed infections within 4 months after surgery, with a significant difference between patients with pre-transplant colonization and non-carriers. Similar results were obtained by Aguado et al. [38], showing that 5.15% of colonized LT recipients developed an infection by ESBL-producing Enterobacterales, compared to 2.4% of non-colonized recipients.

Variables significantly associated with infection in univariate analysis were pre-transplant ESBL fecal carriage, acute liver failure, increased model for end stage liver disease (MELD) score, prolonged hospitalization during the six months before LT, preoperative intensive care stay >48 h, return to surgery, and postoperative acute renal failure. Confirmed independent predictors of infections by ESBL-producing bacteria were pre-transplant fecal carriage, MELD score > 25 and return to surgery.

#### 3.1.2. Colonization/Infection by Carbapenem-Resistant Enterobacterales (CRE)

Emergence of CRE is a major challenge for SOT population, as infections sustained by these microorganisms are associated with increased mortality in different studies.

A multicenter study conducted in SOT recipients in Italy [44] showed a crude incidence rate of Gram-negative bacteria isolation of 2.39 per 1000 recipient-days. The prevalence CRE was 26.5% and *Klebsiella* spp. was the most frequent bacterial species identified (49.1%).

Among LT recipients, several studies analyzed risk factors associated with *Klebsiella pneumoniae* carbapenem-resistant (KPC-Kp). Overall, the principal independent risk factors associated with infections were MELD at LT > 32 [45], hepatocellular carcinoma, Roux-en-Y biliary choledochojejunostomy, and bile leak [46]. Other factors that emerged in different studies were renal-replacement-therapy, prolonged mechanical ventilation (>48 h), hepatitis C virus infection recurrence [47], reintervention, and rejection [45].

An Italian prospective study among 237 LT patients [48] showed a significant difference on KPC-Kp infection rates among patients who were non-colonized, colonized at LT, and colonized after LT (2%, 18.2%, and 46.7%, respectively; *p* < 0.001). KPC-Kp colonization at any time is an independent risk factor for KPC-Kp infection. These results were confirmed in another prospective study analyzing the impact of CRE colonization on risk of infections [45]. The authors showed that CRE colonization at LT and CRE colonization acquired after LT were the strongest risk factors for CRE infection [45]. In this analysis, CRE infection occurred within a median of 31 days after LT (IQR 31-115), with an incidence of 3.05 cases per 10,000 LT-recipient-days.

Similarly, prevalence of CRE infections in KT recipients ranges from 2% and 26% with an in-hospital mortality ranging 33–41%. Risk factors associated with infections are multi-organ transplantation and the use of ureteral stents [48].

In conclusion, among SOT recipients, colonization by GN-MDR plays a very important role in the risk of a subsequent infection. CRE-infection, in turn, has a significant impact on survival. It is, therefore, mandatory to devise strategies to minimize the risk of post-transplant infection and related mortality.

### 3.2. Patients with Hematologic Malignancies

Among patients with hematologic disorders, infections by MDR pathogens have an extremely relevant clinical impact, both for the management of infectious complications, and for the timing of hematological transplant [49]. Hematological cancer patients are a population at risk for MDR infections, due to several conditions—prolonged neutropenia and lack of immune control, risk of intestinal bacterial translocation due to drug toxicity (chemotherapy-induced mucositis, immunosuppression treatment), prolonged antibiotic pressure, and hospitalization.

In a multicentric prospective observational study of 144 hematological patients, 6.5% had an MDR rectal colonization [50] (41% ESBL-producing Enterobacterales, 59% CRE and 6% Vancomicin-resistant Enterococci). BSI occurred in 25.7% of MDR colonized patients, especially during neutropenia (15% ESBL Enterobacterales, CRE 14.1% CR-GN and 11% VRE). Overall mortality at 80 days was 12.5% and 3 month-overall survival was significantly lower among patients with CRE and VRE colonization, compared to those colonized with ESBL-Enterobacterales.

With regard to CRE infections, in a retrospective study (2010–2013) including 52 hematopoietic stem cell transplant (HSCT) centers [51], infections by KPC-Kp were reported in 53.4% of the centers, involving 0.4% of autologous and 2% of allogeneic HSCTs. In 30% of cases, infections followed a previous colonization; the mortality rate of infection was 16% in autologous and 64.4% in allogeneic SCT. The impact of CRE infection on mortality was analyzed in another study involving patients with acute myeloid leukemia, highlighting that patients with CRE colonization had a significant reduction in two years overall survival compared to non-colonized patients [52].

Overall, risk factors associated with MDR infections in hematological patients are identified in—previous exposure to antibiotics, MDR colonization, and allogeneic transplant [53] were found to be the main risk factor for ESBL and CRE infection [54]. In addition, specific risk factors for infections by ESBL-producing bacteria included recent hospitalizations, ICU admission, prolonged duration of hospitalization, and neutropenia [55,56,57].

Given the relevant impact on the mortality of infections sustained by MDR-GN bacteria in hematological patients, MDR bacteria colonization seems to be an extremely important factor contributing to patient outcomes after stem-cell transplant. In 2015, an Italian multidisciplinary group stated that stem cell transplantation might be contraindicated or postponed in patients with a recent history of KPC-Kp infection, and, among colonized patients, transplantation might be delayed to allow for CR-Kp decolonization [49]. This indication was subsequently reconsidered in view of the availability of new drugs to treat MDR. However, MDR colonization still represents a serious issue that needs a concerted action to reduce the impact of infective complications on patient outcomes.

## 4. Strategies for Gut Microbiota Modulation

### 4.1. Oral Antibiotics for MDR Decolonization

MDR-GNB decolonization using oral antibiotics, often referred to as selective digestive decontamination (SDD), consists of administering non-absorbable antimicrobial agents aiming to eradicate yeasts, *Staphylococcus aureus* and (facultative) aerobic Gram-negative bacteria. SDD is mostly performed in the ICU setting, and oral administration of topical non-absorbable antibiotics can be combined with a short initial course of intravenous antibiotics.

Numerous combinations of antibiotics were tested. The agent more often used is oral colistin, administered alone or in combination with oral aminoglycosides (neomycin, amikacin, or tobramycin), erythromycin, rifaximin, or norfloxacin.

Randomized controlled trials set in ICUs with low MDR endemicity [42,58], showed that SDD reduced infections and mortality with limited impact on new resistance selection. However, these studies had major limitations, such as the use of different combinations of antibiotics, the heterogeneity of patient conditions, and ward colonization pressure.

The latest clinical guidelines on decolonization of MDR-GNB published by the European Society of Clinical Microbiology and Infectious Diseases (ESCMID)—European Committee on Infection Control (ECIC) [59] provided a thorough review of the current literature on the subject. A total of 27 studies were analyzed, focusing on five groups of MDR-GNB (namely third-generation cephalosporin-resistant Enterobacterales, carbapenem-resistant Enterobacterales, fluoroquinolone-resistant Enterobacterales, aminoglycoside-resistant Enterobacterales, and carbapenem-resistant *Acinetobacter baumannii*). In conclusion, the panel did not recommend routine decolonization of MDR-GNB carriers, due to a lack of randomized clinical trials with proper sample size and study design assessing the effectiveness and safety.

### 4.2. Fecal Microbiota Transplantation (FMT)

CDI was the first condition in which modulation of gut microbiome was considered a therapeutic option, more than half a century ago. In 1958, Eiseman and colleagues [60] were the pioneers of what has become known as fecal microbiota transplantation (FMT), the administration of feces from a healthy donor to the gastro-intestinal tract of a recipient patient, in order to restore the intestinal microbiome [61]. Since the first experiment, a growing amount of evidence was collected to prove its efficacy—during the last decade, randomized controlled trials, systematic reviews, and meta-analyses showed that FMT can be an effective treatment against CDI [62,63,64,65]. Specifically, FMT is now considered a recommended therapeutic option for patients with multiple recurrences of CDI, when appropriate antibiotic treatments have failed, as per the latest guidelines of the Infectious Diseases Society of America (IDSA) and the Society for Healthcare Epidemiology of America (SHEA) [66].

The evidence of the effectiveness of FMT in treating CDI led to increasing interest in expanding its fields of application. Particularly, since 2013, studies started to focus on the use of FMT as a tool to fight MDR bacteria colonization. In murine models of intestinal colonization with vancomycin-resistant enterococci (VRE), Ubeda et al. [67] proved that fecal transplantation was able to eradicate such colonization, prompting further human studies investigating the potential of FMT in MDR decolonization. Furthermore, Dinh et al. [68] showed how this therapeutic option could be equally effective addressing MDR-GNB with different antibiotic-resistance profile.

Given the burden of MDR infections in individuals with a compromised immune system and the crucial role of MDR bacteria colonization on subsequent development of recurrent infectious episodes, researchers have started focusing on these group (i.e., patients with hematologic diseases and organ transplants recipients) to assess the efficacy and safety of fecal transplantation.

#### 4.2.1. FMT in Solid Organ Transplant Recipients

Patients undergoing SOT are at particular risk of developing infections due to MDR bacteria, given their state of immunosuppression, the frequent hospital stays, and the presence of various comorbidities. The identification of a successful prophylactic approach to deal with MDR bacteria colonization in this population has become a field of growing interest (Table 1). In 2014, Singh et al. [69] performed FMT to address ESBL-producing *E. coli* colonization in a 60-year-old man with end-stage renal disease who underwent two kidney transplantations and transplantectomy due to graft failure. FMT was performed by nasoduodenal tube (NDT), preceded by colon lavage. MDR bacteria eradication was achieved at day 14 after procedure and rectal swab cultures were still negative at 12 weeks follow-up. The only adverse event reported were mild diarrhea and abdominal cramps. After this experience, the same research group carried out a prospective study on the use of FMT against intestinal colonization by ESBL-producing Enterobacterales [70]. Fifteen patients were enrolled, 5 of which (33%) were renal transplant recipients using immunosuppressive drugs. Each patient underwent complete colon lavage with macrogol, one day prior to FMT, which was delivered by NDT. At day 14 post-FMT, MDR bacteria eradication was successful in 6 out of 15 patients (40%). Three of these patients required a second FMT to achieve decolonization—one FMT procedure had a 20% success rate (3/15 patients), while two FMTs had a 43% success rate (3/7 patients). Considering the immunocompromised patient cohort only, one out of 5 participants achieved decolonization (20%); the remaining 4 maintained gut colonization, despite two FMTs. The only adverse events reported following the procedures were mild abdominal discomfort and transient diarrhea (<24 h).

Stripling et al. [71] reported the case of a 33-year-old female with recurrent CDI and VRE infections who underwent heart and kidney transplant. Her immunosuppressant regime consisted of cyclosporine, sirolimus, and prednisone. FMT was delivered via a nasogastric tube, without prior preparation. Loss of VRE fecal dominance was documented at weeks 1 and 3 and at seven months after the procedure. Moreover, no further CDI or VRE infections were reported in the 12 months following FMT. No adverse events were reported following FMT.

Biehl et al. [72] were the first to apply FMT to treat recurrent UTIs caused by MDR-GNB. In their case report, FMT was performed in a 50-year-old female, kidney transplant recipient, with ongoing immunosuppressive treatment (tacrolimus, mycophenolate mofetil, steroid), to address recurrent UTIs from ESBL-producing *E. coli*. FMT was delivered via oral capsules in two consecutive days, without prior bowel lavage. At microbiological follow-up (day 14, 39, and 84 post-FMT), the patient remained culture negative. Furthermore, microbiota analysis revealed a marked decrease in Enterobacterales in urine specimens over time, suggesting an impact of gut microbiota on ecosystems other than the intestinal one, such as vaginal and urinary microbiotas. No adverse events were reported.

Grosen et al. [73] reported the use of FMT for ESBL-producing *K. pneumoniae* recurrent UTIs in a 64-year-old patient, with renal transplant for diabetic nephropathy. The patient had experienced multiple episodes of urosepsis caused by ESBL-producing *K. pneumoniae* during subsequent hospital admissions, requiring carbapenem treatment. Prior to FMT, the patient was also diagnosed with CDI. FMT was delivered via NDT. No adverse events were reported for FMT and the resolution of diarrhea was achieved. Six days later, the patient was re-admitted to the hospital for urosepsis, due to ESBL-producing *K. pneumoniae* and was successfully treated with meropenem. At subsequent follow-up (up to 12 months), no further episodes of UTI occurred and ESBL-producing *K. pneumoniae* was not isolated in urinary or fecal samples collected at outpatient follow-up, 4 and 8 months after FMT.

The results of these first attempts to address MDR bacteria decolonization in SOT recipients are of great interest. Yet, the available data are still limited and there is a wide heterogeneity in procedures, patient selection, and long-term decolonization, thus far. Serious adverse events were not reported during the FMT procedures and the subsequent follow-up. Such encouraging results need to be consolidated through larger and more controlled studies.

#### 4.2.2. FMT in Patients with Hematologic Malignancies

Patients with hematologic malignancies are another population whose survival and quality of life might benefit from MDR bacteria decolonization (Table 2). In 2014, Freedman et al. [74] were the first to report on the use of FMT to eradicate colonization of KPC-Kp. Their patient, a 14-year-old girl with a recent diagnosis of hemophagocytic lymphohistiocytosis treated with corticosteroids and etoposide, developed fever, and KPC-Kp was isolated through blood culture. Despite the use of antibiotics with in vitro activity against the microorganism, the blood cultures remained positive for 5 weeks and the patient developed septic arthritis of one shoulder and both hips and, 10 months after the acute infection, the patient was diagnosed with KPC-Kp osteomyelitis of the right femur. Stool cultures continued to show predominance of the KPC-Kp during the whole period. The patient underwent FMT by NDT, following a 48-h bowel cleanse with polyethylene glycol and pre-medication with proton pump inhibitor (PPI) omeprazole. During the 8 months follow-up after FMT, the patient had KPC-Kp-free stool samples and no recurrence of clinical infection in the 18 months following FMT.

Bilinski et al. [75] used FMT to target MDR bacteria colonization in a 51-years old man with progressive multiple myeloma and severely impaired innate (absolute neutrophil count < 1.0 × 10^9^/L) and acquired (serum IgG concentration 3.83 g/L) immunity. The patient had previously received multiple courses of chemotherapy and three autologous stem cell transplantations. The patient had gut colonization by ESBL-producing *E. coli* and *K. pneumoniae* producing the carbapenemase New Delhi metallo-β-lactamase (NDM). FMT was performed via NDT, after bowel lavage with oral laxative and PPI administration; the only adverse event reported was a single episode of loose stool and transient mild abdominal discomfort. The surveillance rectal swabs collected on day 10 and 26 after FMT tested negative for the MDR bacteria previously cultured. The patient was then lost to follow-up. This case report was followed by a prospective study including 20 patients with blood disorders, excluding severe neutropenic patients (of note, 40% of participants had neutropenia < 1.8 × 10^9^ neutrophils/L) [76]. The MDR bacteria colonizations encompassed in the study were carbapenemase-producing *K. pneumoniae*, ESBL-producing *K. pneumoniae*, and *E. coli*, carbapenemase-producing *P. aeruginosa*, carbapenem-resistant *Enterobacter cloacae*, and VRE. FMT was carried out via NDT, preceded by bowel lavage and PPI usage. The first participants underwent FMT on a single day, while the remaining participants underwent FMT on two consecutive days. At day 30, after the procedure, complete MDR bacteria decolonization was achieved in 15/20 (75%) participants. Notably, among the three patients who received one-day FMTs, no cases of complete MDR bacteria decolonization occurred and only one participant achieved partial decolonization at one month. The mean follow-up period was 187 days (range, 9–482 days); the adverse events reported were either mild and transient gastro-intestinal symptoms, or exacerbations of known conditions. An episode of sepsis following a persisting skin infection caused by carbapenemase-producing *P. aeruginosa* was reported. Another participant with severe chronic graft-versus-host disease died within the first month post-FMT because of septic shock caused by carbapenemase-producing *E. coli*, which was previously eliminated from the gastrointestinal tract. In two participants, recolonization with the same strain of MDR bacteria was observed at about one and six months after decolonization, respectively.

Innes et al. [77] performed FMT to eradicate MDR bacteria (ESBL-producing *E. coli* and carbapenemase-producing *Klebsiella oxytoca*) in a 63 years old man with Philadelphia-positive acute lymphoblastic leukemia who underwent chemotherapy. The patient received gut preparation with four days of oral vancomycin and neomycin (both 500 mg four times daily), iso-osmotic bowel purgatives, PPI, and metoclopramide. FMT was delivered by a nasogastric tube. Following the procedure, the patient experienced mild nausea, loose stools, and transient abdominal discomfort. Eradication of both *K. oxytoca* and ESBL+ *E. coli* was achieved at day 16 post-FMT. Two weeks after the procedure, the patient underwent allogeneic hematopoietic stem cell transplantation (HSCT); the transplant course was complicated by an episode of neutropenic fever, with isolation of a fully sensitive *Enterococcus faecalis* from blood cultures.

Battipaglia et al. [78] studied the use of FMT in patients with hematologic malignancies and gut colonization by MDR bacteria, carrying out a retrospective study focused on individuals pre- and post-allogeneic HSCT.

The microorganisms considered were VRE, carbapenemase-producing Enterobacterales, and carbapenemase-producing-*P. aeruginosa.* Ten patients were studied—4 underwent FMT before HSCT, while 6 patients underwent FMT after HSCT. All patients undergoing FMT after HSCT were still on immunosuppressive therapy at the time of FMT, with only one out of six presenting active grade IV steroid-dependent gut Graft versus Host Disease (GvHD). Overall, six patients were also colonized by ESBL-producing Enterobacterales.

FMT was delivered by enema in all but one patient who had compromised neurological status, due to cerebral toxoplasmosis and was not considered eligible for enema (nasogastric tube administration was used). At the time of the procedure, neutrophil count was >1 × 10^9^/L in all patients but one, who had steroid-resistant GvHD and a neutrophil count of 0.17 × 10^9^/L. Three patients required a second FMT—in one patient, after initial success, VRE was again detectable two months after the first FMT; in the other two patients, a second attempt was made due to failure of the first procedure. After a median follow up of 13 months (range, 4–40) from the procedure, persistent decolonization from the MDR bacteria considered was achieved in six out of ten patients (60%). Among the six patients with concurrent colonization by ESBL-producing Enterobacterales, three (50%) obtained concomitant decolonization. No major adverse events were reported—one patient had constipation during the first five days after FMT, while two had transient diarrhea. Over the three months following FMT, two patients experienced bacteremia without sepsis early after allo-HSCT (blood cultures positive for multi-sensitive *P. aeruginosa* and ESBL-producing *E. coli*), with favorable response to empiric therapy. Despite the use of broad-spectrum antibiotics, cases of MDR bacteria colonization were not observed in these patients.

In a recent case series, Merli et al. [79] studied FMT pre-HSCT in pediatric patients affected by hematologic malignancies and colonized by carbapenemase-producing Enterobacterales and *P. aeruginosa*. The patients median age was 14 years (range, 2–19 years) and three of them had a history of systemic infections by MDR pathogens, subsequent to prior colonization, with a severe clinical course requiring ICU admission. FMT was delivered via a nasogastric tube. At day 7, MDR bacteria eradication was achieved in 4 out of 5 patients (80%). Interestingly, at day 30 post-FMT, the patient that was still colonized achieved decolonization, while the four previously decolonized participants were found to be colonized again by the same pathogen identified before FMT. At the last microbiological follow-up (mean time 56 days, range 28–113), only one out of 5 patients (20%) was still not colonized. Two patients experienced sepsis post-FMT due to the same MDR pathogen colonizing their intestine—one patient five days after HSCT (17 days after FMT) and the other 24 h after FMT. The latter was attributed to the central venous line contamination by the caregiver. Both episodes were resolved with effective antibiotic treatment. No other major adverse events were reported—three patients had mild gastrointestinal disturbances (nausea, bloating, and abdominal discomfort).

## 5. Conclusions and Future Perspective

To date, just little of the complex relationship between intestinal microbiota and MDR colonization is fully understood. In future, the new “-omics” technologies will allow the investigation of the slightest microbiological and genetic changes in the host ecosystems and how they relate to colonization and infection [80]. More importantly, we could achieve a better understanding of the risk factors promoting and sustaining microbiota dysbiosis and leading to subsequent MDR colonization, as well as the factors favoring the switch from the colonization status to local and systemic infection.

Donor selection plays a critical role within the FMT process, requiring a more thorough investigation as well.

First, in order to avoid exposing patients to an additional risk of iatrogenic infections. In June 2019, after DeFilipp et al. [81] described the case of two immunocompromised adults who received FMT and developed invasive infections caused by the ESBL-producing *E. coli*. This led the U.S. Food and Drug Administration to issue a warning addressing the risks of FMT and the need of a thorough donor tool screening, with a wide panel of microbiological and molecular assays. Second, discrepancies in data on the effectiveness of FMT might depend on the heterogeneous choice of stool donors, and on their compatibility with recipients. Being “healthy” is not the only requirement for the donor to possess, in order to perform a successful FMT. There is growing evidence within the various fields of application of FMT (ranging from CDI and MDR decolonization to autoimmune and chronic bowel disease treatment, to cancer immunotherapy [82]) that there exist individuals whose stools are associated with significantly more successful FMT outcomes [83]. These “super-donor” have microbiota characteristics, in terms of composition and capability of modulating the host immune response (e.g., adequate presence of butyrate-producing anaerobic bacteria, such as *Faecalibacterium prausnitzii*, *Roseburia hominis*, *Coprococcus eutactus* [84], and *Barnesiella* spp. [76]), which allow FMT to be more effective. Moreover, the relationship between microbiota composition and decolonization of specific MDR bacteria needs to be further evaluated.

As in every novel field of investigation, FMT needs to be further studied and there are several ongoing trials that will help assess its efficacy and safety on a larger scale (Table 3). Currently, there are fifteen studies ongoing. Most of them have study populations larger than those included in the existing research data. Immunocompromised populations, namely SOT patients, are the target of three studies. Randomized clinical trials are warranted to investigate the efficacy of FMT in treating conditions other than CDI. Finally, it is essential to gather more data on FMT use in immunocompromised hosts, a population that would largely benefit from new approaches to MDR decolonization. 

## Figures and Tables

**Table 1 ijms-21-05619-t001:** Fecal microbiota transplantation (FMT) for multidrug-resistant (MDR) decolonization in solid organ transplant recipients.

Authors	Study Type	Number of Patients	Age	Sex	Multidrug-Resistant Bacteria	Outcome	Route of FMT Administration	PPI	Immunocompromission	Decolonization	Last Available FU	Serious Adverse Events
Singh, 2014 Nederlands	Case report	1	60	M	ESBL *E. coli*	Decolonization	Nasoduodenal tube	No	Multiple kidney TX recipient	Eradication at day 14	Eradication at 12 weeks	Diarrhoea and abdominal cramps
Singh, 2018 Nederland	Prospective monocentric study	15	56 (21–76)	5 M 10 F	ESBL-Enterobacterales	Decolonization	Nasoduodenal tube	No	5 patients were renal transplant recipients using immunosuppressive drugs—1 had ARB eradication	6/15 (40%) at day 14 (7 patients needed 2 FMTs) 3/15 (20%) after first FMT	6/15 at day 28	No major adverse events; temporary loose stools (<24 h)
Stripling, 2015 USA	Case report	1	33	F	VRE	Decolonization	Nasogastric	No	Kidney and heart Tx	No further VRE infections	No VRE infection at 12 months	No adverse events reported
Grosen, 2019 Denmark	Case report	1	64	M	ESBL *K. pneumoniae*	Decolonization	Nasoduodenal tube	N/A	Kidney Tx recipient	Eradication at 4 months	Eradication at 8 months	No adverse event reported
Biehl, 2018 Germany	Case report	1	50	F	ESBL *E. coli*	Decolonization	Oral capsules	N/A	Kidney Tx recipient	Eradication at day 14	Eradication at day 84	No adverse event reported

ESBL: extended spectrum beta-lactamase; VRE: vancomycin resistant enterococcus; ARB: antibiotic-resistant bacteria; Tx: transplant; FMT: fecal microbiota transplantation; PPI: proton pump inhibitor; N/A: not available; FU: follow-up.

**Table 2 ijms-21-05619-t002:** FMT for MDR decolonization in patients with hematologic malignancies.

Authors	Study Type	Number of Patients	Age	Sex	Multidrug-Resistant Bacteria	Outcome	Route of FMT Administration	PPI	Immunocompromission	Decolonization	Last Available FU	Serious Adverse Events
Bilinski, 2017 Poland	Prospective monocentric study	20	51 (22–77)	14 M 6 F	CRE, VRE, ESBL, CR *P. aeruginosa*	Decolonization	Nasoduodenal tube	Yes	All patients with blood disorders 8 patients with ANC 0.5–1.8 x 10^9^/L	15/20 (75%) at day 30	13/14 at 6 months	Vomiting 1 diarrhea 20 Abdominal pain 2 Ileus 2
Innes, 2017 United Kingdom	Case report	1	36	M	CRE; ESBL *E. coli*	Decolonization	Nasogastric tube	Yes	Philadelphia-positive acute lymphoblastic leukaemia	Eradicated at day 16	No further infection at 12 months	Mild nausea, loose stool and abdominal discomfort (<24 h). Enterococcus faecalis bacteremia
Freedman, 2014 USA	Case report	1	14	F	CR *K. pneumoniae*	Decolonization	Nasoduodenal tube	Yes	Hemophagocytic lymphohistiocytosis	Eradication	Eradication at 18 months	No adverse events reported
Bilinski, 2016 Poland	Case report	1	51	M	*CR K. Pneumoniae*; ESBL E. coli	Decolonization	Nasoduodenal tube	Yes	Progressive multiple myeloma and severely impaired innate and acquired immunity	Eradication at day 10	Eradication at 1 month (lost at FU)	Loose stool and transient mild abdominal discomfort
Merli, 2020 Italy	Case series	5	14 (2–18)	4 M 1 F	CRE; CR *P. aeruginosa*	Decolonization	Nasoduodenal tube	No	Before HSCT	4/5 (80%) at day 7	1/5 (20%) at day 30 (4 patients were colonized again by the same bacteria)	2 episodes of sepsis (one attributed to contamination of central venous line)
Battipaglia, 2019 France	Restrospective monocentric study	10	48 (16–64)	4 M 6 F	CRE, CR *P. aeruginosa*, VRE	Decolonization	Enema (9); nasogastric tube (1)	No	4 patients before allo-HSCT; 6 patients after allo-HSCT All patients but one had ANC > 1 × 10^9^/L (patient with steroid-resistant GvHD ANC 0.17 × 10^9^/L)	7/10 (70%)	6/10 (60%) at follow up (median 13 months, range, 4–40 months)	No major adverse events Costipation 1 diarrhea 2 In the 90 days following FMT: 2 bacteriemia

ESBL: extended spectrum beta-lactamase; CR: carbapenem resistant; HSCT: hematopoietic stem cell transplantation; VRE: vancomycin resistant enterococcus; FMT: fecal microbiota transplantation; GvHD: Graft versus Host disease; CRE: carbapenem resistant *Enterobacteriales*; PPI: proton pump inhibitors; ANC: absolute neutrophil count; FU: follow-up.

**Table 3 ijms-21-05619-t003:** Ongoing studies on FMT for MDR decolonization ((Clinicaltrials.gov), last access 30th June 2020).

Clinical Trial Number	Study Design	Clinical Setting	Outcomes
FMT for multidrug resistant organism reversal NCT 02312986	Prospective pilot study Enrolment prevision: 20 patients	Outpatients with previous infections with MDR	Primary: Safety of FMT (12 months after FMT) Secondary: MDR infections status post FMT (month 1, 6 and 12 after FMT)
Biotherapy for MRSA Enterocolitis NCT 02390622	Pilot study Enrolment prevision: 10 patients	Inpatients with MRSA enterocolitis	Faecal microbiota diversity (1 month 1 after FMT)
FMT as a strategy to eradicate resistant organisms NCT 02543866	Prospective pilot study Enrolment prevision: 20 patients	Paediatric outpatients with a history of Extended Spectrum resistant (ESC-R) Enterobacteriaceae colonization	Primary: Safety and tolerability of FMT Secondary: Efficacy of FMT (2 days, week 2, 4, 8, month 6 and 12 after FMT)
FMT for MDRO Colonization in solid organ Transplant NCT 02816437	Pilot feasibility study Enrolment prevision: 20 patients	Inpatient solid organ transplant recipients colonized by CRE, VRE, or CRP without active infections	Primary: Adverse events Secondary: Rate of MDR colonization Rate of recurrent MDR colonization
A trial of encapsulated fecal microbiota for Vancomycine resistant Enterococcus Decolonization NCT03063437	Phase II Randomized, Double Blind, Placebo-controlled, Parallel Group Trial	Outpatients or inpatient with VRE colonization without active infection	Primary: -VRE decolonization (10 days after randomization) -Safety Secondary: -VRE infection within 4 weeks -Other MDR colonization and infection in 4 weeks after FMT
Fecal Microbiota Transplantation for eradication of CRE NCT03167398	Prospective mono-centre study Enrolment prevision: 60 patients	Inpatients colonized with CRE	Primary: CRE eradication in rectal stool samples at month 1 (3 consecutive negative rectal sample)
Fecal Microbiota Transplantation for MDRO UTI NCT 03367910	Prospective study Enrolment prevision: 60 patients	Outpatient status	Primary: ● FMT safety Secondary: ● risk of recurrent UTI ● MDR colonization
FMT for MDRO Colonization After Infection in Renal Transplant Recipients (PREMIX) NCT 02922816	Interventional (clinical trial) Enrolment prevision: 20 patients	Adults who have undergone renal transplantation and have a history of infection with the Target Multidrug Resistant Organisms (MDRO)	Primary outcomes: The safety and feasibility, measured by comparing the number of adverse events after Day 1 of each cycle as compared to baseline
Fecal Transplant, a Hope to Eradicate Colonization of Patient Harboring eXtreme Drug Resistant Bacteria? (FEDEX) NCT03029078	Single group open label trial Enrolment prevision: 50 patients	Patients harboring XDR bacteria in their digestive tract	Primary outcomes: negativization of digestive tract colonization [Time Frame: At one week, 2 weeks, one month up to 6 months] Secondary outcomes: Metagenomics profile between donor and recipient [Time Frame: At one week, 2 weeks, one month up to 3 months ]
Fecal Microbiota Transplantation for Carbapenem-resistant Enterobacteriaceae NCT04146337	Randomized open label trial Enrolment prevision: 60 patients	CRE carriers	Primary outcomes: CRE eradication [Time Frame: 28 days] Secondary outcomes: CRE eradication rates at day 7, day 14, and 3 & 6 months; mortality; bacteriemia; CRE infection; hospitalization days; adverse events
Effectiveness of Fecal Flora Alteration for Eradication of Carbapenemase-producing Enterobacteriaceae Colonization (EFFECT-CPE) NCT03802461	Randomized open label trial Enrolment prevision: 40 patients	Asymptomatic adult patients intestinally colonized with CPE	Primary outcomes: Incidence of intestinal colonization of patients with CPE 3 months after intervention. [Time Frame: 3 months ]; Randomization rate in study [Time Frame: 12 months ]; Proportion of patients retained in study for up to 6 months [Time Frame: 6 months]
Fecal Microbiota Transplantation for CRE/VRE NCT03479710	Non randomized open label trial Enrolment prevision: 40 patients	Patients with two or more stool or rectal swab positive for CRE or VRE at least one week apart	Primary outcomes: Intestinal colonization of CRE/VRE [Time Frame: 2 weeks to 12 months] Secondary outcomes: adverse events; changes in intestinal microbiota
Fecal Transplant for MDRO Decolonization NCT04181112	Randomized open label trial Enrolment prevision: 90 patients	Renal transplant patients positive for one of the target MDRO by rectal or stool culture tests	Primary outcomes: The elimination of the target multi-drug resistant organism (MDRO), using culture and molecular test-based screening of recipient stool, at both the 14 and 30 days post-FMT. [Time Frame: 3 years] Secondary outcomes: adverse events; recolonizations
Phase II Trial of Fecal Microbiota Transplant (FMT) for VRE and CRE Patients NCT 03643887	Randomized blinded trial Enrolment prevision: 90 patients	Adults who have had two consecutive positive stool cultures for VRE or CRE	Primary outcomes: compare incidence of VRE/CRE decolonization between placebo and FMT Secondary outcomes: adverse events/serious adverse events through Day 10, Week 4, and Month 6 following randomization
Decolonization of Gram-negative Multi-resistant Organisms (MDRO) With Donor Microbiota (FMT) (DEKODON) NCT04188743	Randomized blinded trial Enrolment prevision: 150 patients	Patients with at least 2 consecutive confirmations of MDRO colonization in faeces	Primary outcomes: Number of participants with decolonization success/failure [Time Frame: 1 month after treatment] Secondary outcomes: adverse events; treatment effect on microbial community in participants; treatment tolerability (questionnaire)

MDRO: multi-drug resistant organism; MRSA: Methicillin-resistant Staphylococcus aureus; CRE: Carbapenem-Resistant Enterobacteriaceae; VRE: Vancomycin-Resistant Enterococci; CRP: Carbapenem-Resistant Pseudomonas; NDM: New Delhi metallo-beta-lactamase; MBL: metallo-beta-lactamase; ESBL: Extended-spectrum-beta-lactamase; CDI: C. difficile infection; and UTI: urinary tract infections.

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
