# Peer review of "Multidrug-Resistant Gram-Negative Bacteria Decolonization in Immunocompromised Patients: A Focus on Fecal Microbiota Transplantation"

_ijms, 2020, doi:10.3390/ijms21165619_

Round 1

Reviewer 1 Report

Multidrug-Resistant Gram Negative Bacteria Decolonization in Immunocompromised Patients : A Focus on FMT is a Review on MDR or XDR colonization in SOT or Hematological patient.

Authors organized their review in two part after a wide introduction on both AMR burden and microbiota.

In the part 3 which is for me the first part of the review they provide MDR-GNB overview in both population (SOT and Hematologic malignacy patient). This part is well documented but classical.

In part 4 they review two different approaches to eradicate colonization :SDD and FMT.

Due to the little data available at this time articles included in this review are mostly case reports or FMT performed for another eradication with as a by-product the eradication of a MDR bacteria.

But this part is well written and they provide a good overview lacking just one or two references .

Discussion point the secondary effect of FMT but did not give data on positive side effect as point by Zyvogel's team on oncologic therapeutic response impact of FMT.

The table will be highly improved with an added column showing the log term decolonization persitence when avvailable. In fact when FMT was performed is the patient still eradicate at D30, D45, D60... This is a main concern to promote FMT.

Lastly Table 2 provide list of study still ongoing on FMT in these population.

Its place in this manuscript should be discussed and may be formated to provide less information on each study but synthetized studies by intervention, population outcome.

Minor :

Liquor could be replace with fluid

Authors could add this reference

- J Hosp Infect. 2018 Aug;99(4):481-486. doi: 10.1016/j.jhin.2018.02.018. Epub 2018 Mar 2. PMID: 29477634 Clinical Trial.

Author Response

Thank you for the opportunity to revise our manuscript, previously entitled “Multidrug-Resistant Gram-Negative Bacteria Decolonization in Immunocompromised Patients: A Focus on Fecal Microbiota Transplantation”. We are grateful for the extensive modifications proposed by the reviewers, and we followed their suggestions in this revision.

Reviewer 1:

Point 1: Multidrug-Resistant Gram-Negative Bacteria Decolonization in Immunocompromised Patients: A Focus on FMT is a Review on MDR or XDR colonization in SOT or Hematological patient.

Authors organized their review in two part after a wide introduction on both AMR burden and microbiota.

In the part 3 which is for me the first part of the review they provide MDR-GNB overview in both population (SOT and Hematologic malignacy patient). This part is well documented but classical.

Answer: We are grateful to the reviewer for this comment. We organized this section in a “classical” description of all emerging data in specific immunocompromised patients (SOT and Haematologic malignancy patients) to give the readers the necessary elements to understand the rationale of decolonization interventions in these settings.

Point 2: In part 4 they review two different approaches to eradicate colonization: SDD and FMT.

Due to the little data available at this time articles included in this review are mostly case reports or FMT performed for another eradication with as a by-product the eradication of a MDR bacteria.

But this part is well written and they provide a good overview lacking just one or two references.

Answer: We agree with the reviewer’s comment about the limited data available for clinical use of FMT as MDR-bacteria decolonization strategy. We added one reference we missed in the first version of the paper (see Ref N.69)

Point 3: Discussion point the secondary effect of FMT but did not give data on positive side effect as point by Zytvogel's team on oncologic therapeutic response impact of FMT.

Answer: We thank the reviewer for this very important comment. We mentioned in the manuscript the future potential application of FMT as cancer immunotherapy: “ranging from CDI and MDR decolonization to autoimmune and chronic bowel disease treatment to cancer immunotherapy (line 491) and we added the suggested reference (see Ref N. 83).

Point 4: The table will be highly improved with an added column showing the long-term decolonization persistence when available. In fact, when FMT was performed is the patient still eradicate at D30, D45, D60... This is a main concern to promote FMT.

Answer: We thank the reviewer for this suggestion. We implemented TABLE 1 adding the column “LAST AVAILABLE FOLLOW UP” specifying the duration of eradication.

This issue is briefly mention in the manuscript “Yet, available data are still limited and there is a wide heterogeneity in procedures, patient selection and in long term decolonization so far” (Line 380-381)

Point 5: Lastly Table 2 provide list of study still ongoing on FMT in these population.

Its place in this manuscript should be discussed and may be formatted to provide less information on each study but synthetized studies by intervention, population outcome.

Answer: We thank the reviewer for his helpful comment. We edited TABLE 3 keeping information on study design, clinical setting and outcome. We mentioned in the manuscript principal highlights of ongoing studies “So far, fifteen study are ongoing. Most of them have study populations larger than those included in existing research data. Immunocompromised population, namely SOT patients, will be the target of three studies.” (Line 499-502)

Minor:

Liquor could be replaced with fluid

Answer: the term “liquor” has been changed into “cerebrospinal fluid” (Line 62)

Authors could add this reference

- J Hosp Infect. 2018 Aug;99(4):481-486. doi: 10.1016/j.jhin.2018.02.018. Epub 2018 Mar 2. PMID: 29477634 Clinical Trial.

Answer: we thank for this suggestion. We added in the manuscript the indicated article (ref 69), commenting “Furthermore, Dinh et al. showed how this therapeutic option could be equally effective addressing MDR-GNB with different antibiotic-resistance profile.” (Line 329-330)

We appreciate the insightful revisions and hope that our manuscript can now be considered for publication in the International Journal of Molecular Sciences.

Sincerely,

Emanuele Palomba, MD

Laura Alagna, MD

Reviewer 2 Report

The article is wonderfully woven, and the topic lies well within the scope of the journal. The review deals with the prospects of fecal microbiota transplantation and the avenues of their possible use in immunosuppressed patients especially those with blood malignancies. The authors are to be applauded for using a lot of unique terms. The manuscript however does require extensive English grammar review. It also needs to be simplified in some areas to improve overall clarity.

Abstract:

  1. Line # 18: Please rephrase to use appropriate separator as semicolons.
  2. Line #19: Intestinal microbiota dysbiosis refers to the imbalance caused “because” of certain taxa populating, please rephrase to accommodate this change.

Review:

  1. Lines #38-39: redundant use of the term “bacteria”.
  2. Line #41-42: Please correct for grammar.
  3. Line # 43: Please check again if Enterobacteriaceae is a family under the order Enterobacterales or just a new name for it.
  4. Line #47: Please correct for grammar.
  5. Line #51: Liquor, please use a commonly used scientific term instead
  6. Line #57: Please correct for grammar.
  7. Line # 60: Like coli
  8. Line # 62: Please clarify what methods were used or how was regular surveillance done.
  9. Line # 69-70: Please substantiate statistical data with appropriate references.
  10. Line # 76: Please rephrase to imply same individuals.
  11. Line #90: Biliary acids or Bile acids
  12. Line #153: Co-housing or administering, please specify
  13. Line #183: repetition of same meaning words.
  14. Line #189: Incomplete sentence.
  15. Line #218-221: Please refer the study either by the name of the researcher or the research. It can be specified what population was researched upon.
  16. Line #233: incomplete sentence
  17. Line 306-314: Clarity needed.
  18. Line # 317 ; please check if this expression fits here “already more than half a century ago“.
  19. Line#391: “culture”

Author Response

Thank you for the opportunity to revise our manuscript, previously entitled “Multidrug-Resistant Gram-Negative Bacteria Decolonization in Immunocompromised Patients: A Focus on Fecal Microbiota Transplantation”. We are grateful for the extensive modifications proposed by the reviewers, and we followed their suggestions in this revision.

Reviewer 2

Point 1: The article is wonderfully woven, and the topic lies well within the scope of the journal. The review deals with the prospects of fecal microbiota transplantation and the avenues of their possible use in immunosuppressed patients especially those with blood malignancies. The authors are to be applauded for using a lot of unique terms. The manuscript however does require extensive English grammar review. It also needs to be simplified in some areas to improve overall clarity.

Answer: We thank the reviewer for the extensive manuscript editing. We appreciated his comment about English grammar review, and we provided a new grammar revision of the manuscript.

Abstract:

Line # 18: Please rephrase to use appropriate separator as semicolons.

Answer “Antimicrobial resistance is an important issue for global health, and it is even a bigger threat for immunocompromised patients such as solid organ and hematological transplant recipients.” Should be rephrased as “Antimicrobial resistance is an important issue for global health; in immunocompromised patients, such as solid organ and hematological transplant recipients, it becomes even a bigger threat.”

Line #19: Intestinal microbiota dysbiosis refers to the imbalance caused “because” of certain taxa populating, please rephrase to accommodate this change.

Answer:  we are thankful for this clarification. We rephrased into. “A growing pile of studies has linked the imbalance caused by the dominance of certain taxa populating the gut, also known as intestinal microbiota dysbiosis, to an increased risk of MDR bacteria colonization”

Review:

Lines #38-39: redundant use of the term “bacteria”.

Answer: “bacteria” was replaced with “microorganism”

Line #41-42: Please correct for grammar.

Answer: The paragraph “In this review we will focus on MDR gram-negative bacteria (MDR-GNB) that have a significant impact in immunocompromised patients and for which different strategies and management methods are long debated” have been modified as … “In this review we will focus on MDR gram-negative bacteria (MDR-GNB), because of their significance in immunocompromised patients, exploring the strategies and management methods used to address the issue of colonization and subsequent infection.” … (Line 39-41)

Line # 43: Please check again if Enterobacteriaceae is a family under the order Enterobacterales or just a new name for it.

Answer: We thank the reviewer for this comment. We agree with his revision that Enterobacteriaceae is a family under the order Enterobacterales, according the new reclassification of Enterobacteriaceae stated by Adelou M et al (doi:10.1099/ijsem.0.001485. Epub 2016 Sep 11). (Line 42)

Line #47: Please correct for grammar.

Answer: “Among those bacteria, a major mechanisms of AMR are the production …” have been rephrased as “Among those bacteria, a major mechanism of AMR is.”(Line 46)

Line #51: Liquor, please use a commonly used scientific term instead

Answer: The term “liquor” has been changed into “cerebrospinal fluid” (Line 62)

Line #57: Please correct for grammar.

Answer: “Italy is one of the European country with the highest percentage of AMR.” have been rectified as “Italy is one of the European countries with the highest percentage of AMR” (Line 61)

Line # 60: Like coli

Answer: “Likewise” have been replaced by “Like” (Line 64)

Line # 62: Please clarify what methods were used or how was regular surveillance done.

Answer: Thanks for the comment.

We rephrased adding this paragraph … “Data on AMR in Europe are reported by the European Antimicrobial Resistance Surveillance Network (EARS-Net,www.ecdc.europa.eu/en/about-us/partnerships-and-networks/disease-and-laboratory-networks/ears-net). EARS-Net collect results from routine antimicrobial susceptibility testing of invasive isolates (blood and cerebrospinal fluid cultures), from clinical laboratories in each of the 30 European countries involved. The use of EUCAST clinical breakpoints is encouraged.” (Lines 48-53).

In addition, we replaced data, updating with more recent EARS-Net report (2018 instead of 2017).  We replaced Ref N. 3 (European Centre for Disease Prevention and Control External quality assessment of laboratory performance: European Antimicrobial Resistance Surveillance Network (EARSNet), 2017.; 2018; ISBN 978-92-9498-282-7) with the following reference “SURVEILLANCE REPORT. Surveillance of antimicrobial resistance in Europe 2018”.

Line # 69-70: Please substantiate statistical data with appropriate references.

Answer: we added Ref N.3 “SURVEILLANCE REPORT. Surveillance of antimicrobial resistance in Europe 2018”

Line # 76: Please rephrase to imply same individuals.

Answer: The phrase “the relative proportions of each taxon vary considerably between individuals and even within individuals during their lifetime” have been modified as “the relative proportions of each taxon vary considerably between individuals and even within the same individual during his lifetime.” (Line 79-80)

Line #90: Biliary acids or Bile acids

Answer: Biliary acids” has been replaced into “Bile acids”

Line #153: Co-housing or administering, please specify

Answer: This sentence refers to a recent manuscript published by Fay and colleagues in 2019 (see Ref N.29) in which they examined sepsis in genetically identical, age-matched, gender-matched mice, subjected to cecal ligation and puncture (CLP), the most commonly used preclinical model of sepsis. This experimental design took advantage of the fact that mice from different vendors have different microbiomes based on their origin yet are otherwise identical, and these differences can directly impact the host response to infection and inflammation. To determine whether the microbiome is directly responsible for the results generated, unmanipulated mice from different vendors were then cohoused in the same cage with access to the same chow and water for 3 weeks, after which they developed a common microbiome and were then subjected to CLP Thus, co-housing (and administration of the same chow and water) was an established part of their study design to determine the impact of microbiome on survival and immunophenotype.

Line #183: repetition of same meaning words.

Answer: we replaced the sentence “…threaten success of transplantation “as “threaten the change of graft survival”  (Line 185)

Line #189: Incomplete sentence.

Answer: the sentence has been completed as “Different studies aimed to analyze risk factors for MDR colonization and or infections and appropriate management of colonized patients among candidates to SOTs is still a debated topic. “ (Line 191-192)

Line #218-221: Please refer the study either by the name of the researcher or the research. It can be specified what population was researched upon.

Answer: we added the researcher name in the sentence, we specified the target population (liver transplant patients: “ Bert et al (Ref N. 44) analyzed 710 LT patients, reporting that …”; “ Aguado et al (Ref N.39). Line 218-221

Line #233: incomplete sentence

Answer: we completed the sentence as “ A multicenter study conducted in SOT recipients in Italy showed a crude incidence rate of Gram-negative bacteria isolation of 2.39 per 1000 recipient-days.” (Line 232-233)

Line 306-314: Clarity needed.

Answer: The paragraph has been rephrased as “The latest clinical guidelines on decolonization of MDR-GNB published by the European Society of Clinical Microbiology and Infectious Diseases (ESCMID) - European Committee on Infection Control (ECIC) provided a thorough review of the current literature on the subject. A total of 27 studies were analyzed, focusing on five groups of MDR-GNB (namely third-generation cephalosporin-resistant Enterobacterales, carbapenem-resistant Enterobacterales, fluoroquinolone-resistant Enterobacterales, aminoglycoside-resistant Enterobacterales and carbapenem-resistant Acinetobacter baumannii). In conclusion the panel did not recommend routine decolonization of MDR-GNB carriers, due to a lack of randomized clinical trials with proper sample size and study design assessing its effectiveness and safety.” (Line 303-311)

Line # 317 ; please check if this expression fits here “already more than half a century ago“.

Answer:  the sentence “already more than half a century ago” has been rephrased as “more than half a century ago. In 1958…”(Line 314)

Line#391: “culture”

Answer:“ blood colture” has been replaced  by “ blood culture”. (Line 389)

We appreciate the insightful revisions and hope that our manuscript can now be considered for publication in the International Journal of Molecular Sciences.

Sincerely,

Emanuele Palomba, MD

Laura Alagna, MD

Round 2

Reviewer 1 Report

The manuscript in is reviewed form is improved.

Author's answer point point to all my concerns

Their manuscript is well written and provide a good overview of the subject.

To my point of view this manuscript is now suitable for publication.